# Association of Serum Levels of Plasticizers Compounds, Phthalates and Bisphenols, in Patients and Survivors of Breast Cancer: A Real Connection?

**DOI:** 10.3390/ijerph19138040

**Published:** 2022-06-30

**Authors:** Mariana Segovia-Mendoza, Margarita Isabel Palacios-Arreola, Luz María Monroy-Escamilla, Alexandra Estela Soto-Piña, Karen Elizabeth Nava-Castro, Yizel Becerril-Alarcón, Roberto Camacho-Beiza, David Eduardo Aguirre-Quezada, Elías Cardoso-Peña, Omar Amador-Muñoz, José de Jesús Garduño-García, Jorge Morales-Montor

**Affiliations:** 1Departamento de Farmacología, Facultad de Medicina, Universidad Nacional Autónoma de México, Ciudad de Mexico 04510, Mexico; 2Grupo de Especiación Química de Aerosoles Orgánicos Atmosféricos, Instituto de Ciencias de la Atmósfera y Cambio Climático, Universidad Nacional Autónoma de México, Ciudad de Mexico 04510, Mexico; mi.palacios.arreola@gmail.com (M.I.P.-A.); oam@atmosfera.unam.mx (O.A.-M.); 3Centro Nacional Hospital 20 de Noviembre, ISSTE, Col del Valle Sur, Ciudad de Mexico 03100, Mexico; escamilla_lm@issste.gob.mx; 4Facultad de Medicina, Universidad Autónoma del Estado de México, Toluca 50000, Mexico; aesotop@uaemex.mx (A.E.S.-P.); ybecerrila048@alumno.uaemex.mx (Y.B.-A.); camacho_beiza@hotmail.com (R.C.-B.); elias.cardosope@imss.gob.mx (E.C.-P.); jjgardunog@uaemex.mx (J.d.J.G.-G.); 5Grupo de Biología y Química Ambientales, Departamento de Ciencias Ambientales, Instituto de Ciencias de la Atmósfera y Cambio Climático, Universidad Nacional Autónoma de México, Ciudad de Mexico 04510, Mexico; karlenc@atmosfera.unam.mx; 6Unidad Médica Especializada para la Detección y Diagnóstico de Cáncer de Mama, Instituto de Salud del Estado de México, Toluca 51760, Mexico; deaqsol@hotmail.com; 7Unidad de Medicina Familiar 220, Instituto Mexicano del Seguro Social, Toluca 50070, Mexico; 8Hospital Regional 251, Instituto Mexicano del Seguro Social, Toluca 50070, Mexico; 9Departamento de Inmunología, Instituto de Investigaciones Biomédicas, Universidad Nacional Autónoma de México, Ciudad de Mexico 04510, Mexico

**Keywords:** breast cancer, serum levels, phthalates, bisphenols, endocrine-disrupting compounds

## Abstract

Phthalates and bisphenols are ubiquitous environmental pollutants with the ability to perturb different systems. Specifically, they can alter the endocrine system, and this is why they are also known as endocrine-disrupting compounds (EDCs). Interestingly, they are related to the development and progression of breast cancer (BC), but the threshold concentrations at which they trigger that are not well established. Objectives: The aim of this study was to compare the concentration measures of parent EDCs in three groups of women (without BC, with BC, and BC survivors) from two urban populations in Mexico, to establish a possible association between EDCs and this disease. We consider the measure of the parent compounds would reflect the individual’s exposure. Methods: The levels of di-ethyl-hexyl-phthalate (DEHP), butyl-benzyl-phthalate (BBP), di-n-butyl phthalate (DBP) and di-ethyl-phthalate (DEP), bisphenol A (BPA) and bisphenol S (BPS) were determined by gas chromatograph-mass spectrometry in 102 subjects, including 37 women without any pathological disease, 46 patients with BC and 19 women survivals of BC of Mexico and Toluca City. Results: All phthalates were detected in 100% of women, two of them were significantly higher in patients with different BC subtypes in Mexico City. Differential increases were observed mainly in the serum concentration of phthalates in women with BC compared to women without disease between Mexico and Toluca City. In addition, when performing an analysis of the concentrations of phthalates by molecular type of BC, DEP and BBP were found mainly in aggressive and poorly differentiated types of BC. It should be noted that female BC survivors treated with anti-hormonal therapy showed lower levels of BBP than patients with BC. BPA and BPS were found in most samples from Mexico City. However, BPS was undetectable in women from Toluca City. Discussion: The results of our study support the hypothesis of a positive association between exposure to phthalates and BC incidence.

## 1. Introduction

The exacerbated prevalence of BC has critical detrimental effects on the health of the female population around the world [1]. For some years, this disease has been mainly classified into three molecular subtypes based on the expression of the estrogen receptor (ER), progesterone receptor (PR), and epidermal growth factor receptor type II (HER2). According to these molecular markers, the widely recognized subtypes of BC in pathological practice are luminal or ER+/PR+, HER2+, or triple-negative (TN) [2]. Among the different molecular types of this pathology, the luminal one is the most frequent in women worldwide, with a better prognosis in comparison to the two other most aggressive subtypes [3]. Multiple efforts have emerged for its early detection and targeted treatment; however, the incidence of BC is still increasing. Different studies have demonstrated that several risk factors including reproductive, hormonal exposure, epigenetic changes, genetic history, lifestyle, and even exposure to different pollutants, have nowadays contributed to the greater incidence of BC, especially in the ER subtype [4,5]. Phthalates and bisphenols are synthetic compounds considered ubiquitous pollutants present in several plastic devices, food containers, and personal and health products [6,7]. Worldwide, their presence has expanded into different environmental contexts, including food, air, and water, representing a meaningful health issue [5]. Both phthalates and bisphenols have demonstrated a positive correlation with BC development in animal models [8,9,10,11] and with the proliferation of BC cells [12,13,14,15]. Phthalates and bisphenols are implicated in activating different cell signaling pathways through the stimulation of different steroid receptors, especially the ER, characteristic for which they have been called endocrine-disrupting compounds (EDCs) [16,17]. Furthermore, these compounds are implicated in the stimulation of ER-negative BC cells proliferation, including HER2+ and TN BC cells, through a mechanism associated with the activation of different growth factor receptors and mitogenic signaling pathways [11,14,15]. Of note, exposure to both classes of plastic compounds, bisphenols and phthalates, during critical periods of life can predispose to disturbing health effects at multiple levels, including the development of BC [5]. Interestingly, it has been demonstrated that different metabolites of BPA can also interact with different human estrogen receptors, causing delayment in development, deformations, and arrhythmias. This was demonstrated by docking studies and corroborated in an embryo zebrafish model [18]. In addition, different intermediates products of EDCs can be formed through chemical reactions with the aim to obtain an inactive form of them. Fenton process, where OH radicals react with organic analytes, has been widely described for BPA, obtaining p-diphenol, phenol, and isopropyl phenol [19]. which could also interact with different targets but at the same time could limit the EDCs measurement in a complete landscape.

Facts suggest that the high intake of both bisphenols and phthalates in adulthood could be responsible for the high incidence of BC and different health perturbances; but also, in a retrospective context, this predisposition could be inherited through the use of different plastic devices and the accumulation and biological actions of their metabolites in the perinatal stage i.e., mother–infant contact, as we had previously postulated [5]

Due to the remarkable effects of phthalates and bisphenols reported in in vitro and in vivo studies, multiple analyses have been made at the translational level. Some research groups have examined the urinary and serum concentrations of several metabolites of phthalates [20,21,22,23,24,25,26] and bisphenols [27,28,29] in patients with BC to draw a relationship between their appearance and their effects in people with this disease. The results are still not conclusive. Among the main phthalate metabolites associated with the incidence of BC are the monoethyl phthalate (MEP) [20], mono-(2-ethylhexyl) phthalate (MEHP)20, mono-(2-ethyl-5-carboxypentyl) phthalate (MECPP) [26], which are metabolites derived from DEP and DEHP.

Normally, after humans or other mammals ingest the parent form of phthalates and bisphenols, the pollutants undergo at least two metabolic changes, including hydrolysis and conjugation in the liver. Finally, they are excreted through bile or urine with a half-life of approximately 5 h [30,31,32]. Of note, the half-life of phthalates depends on their molecular weight. In this way, compounds with low molecular weight (DEP and DBP) are hydrolyzed, they are further converted into oxidative metabolites through a multistep pathway to monoesters and excreted, while phthalates with high molecular weight (DEHP and BBP) are also first hydrolyzed and then metabolized [33,34]. In the case of BPA or other bisphenols, they undergo glucuronidation to be excreted in the urine [35,36]. Based on their metabolism, different oxidative or glucuronide forms of phthalates and bisphenols might be routinely identified in the urine. Furthermore, they can be found in other fluids such as semen, breast milk, and amniotic fluid, among others. The concentrations in which these compounds have been found are in the ng/mL range [34,37,38,39]. The role of bisphenols in BC is not clear because on one side they can increase cell proliferation of BC cells, but some investigations reveal a lack of relationship between them and the development of BC or its progression and maintenance [26,27,40]. Structural synthetic analogs of BPA such as BPS have emerged to replace it; however, there is not enough evidence about the relationship between BPS serum or urine levels and the progression of BC [41].

Of note, to date, there are no studies about the quantitation of phthalates and bisphenols as parent compounds in the serum of patients with BC around the world. Considering that these contaminants are found in many sources and that both humans and animals are constantly exposed to these compounds, we decided to evaluate the serum concentrations of the parent compounds of four phthalates and two bisphenols in two industrialized cities of Mexico that have similar elevation with respect to sea level. In addition, both Toluca City and Mexico City are contiguous places and are characterized by high levels of air pollution due to their industrial activity. In the case of phthalates, two of them correspond to compounds with high molecular weight, found mainly in plastic devices; the other two correspond to low molecular weight, mainly found in cosmetic and personal use products. We measured the BPA and one of its main analogs, BPS, concerning bisphenols.

## 2. Materials and Methods

### 2.1. Patients/Study Population

Serum from different patients with BC was collected from Hospital 20 de Noviembre (Mexico City), and the Medical Specialties Unit for the Detection and Diagnosis of Breast Cancer (UNEME-DEDICAM) of Health Institute of the State of Mexico (ISEM) Toluca City. Samples of survivors of BC were taken to the General Regional Hospital 251 of the Mexican Institute of Social Security (Toluca, Estado de México). Subjects without any declared pathology were matched by age and place of residence. Of note, Mexico City is considered a highly populated and crowded place. In the same manner, Toluca City is considered a significant industrialized city neighbor to Mexico City. We gathered 46 patients from Mexico City and Toluca City with histopathological confirmation of BC. Moreover, we included in this study 19 women who survived BC after 5 years free of recurrence, this population was named as survivors of BC. The age range of the patients was between 40–65 years. Different clinicopathological data of the BC group (age, histological type, molecular classification, and stage were collected, Table 1).

On the other hand, the controls consisted of 13 (Mexico City) and 24 (Toluca) women without any declared pathology.

### 2.2. Collection of Breast Cancer Samples

For this study, all subjects gave their informed consent for inclusion before they participated in the study. The study was conducted in accordance with the approved Human Research Ethics Committee from the Internal Review Board of the Biomedical Research Institute, UNAM, and the Ethics Committee of the Medical Sciences Research Center of Autonomous Mexico State University (Project number: 2018/10) as well as the authorization of the Institute of Health of the State of Mexico Academic Coordination (217B50025/888/19) of Toluca City. The patients provided their written informed consent to participate in this study. Serum samples of control groups and different cancer patients belonging to Hospital 20 de Noviembre, Mexico City, Hospital General Regional 251 del Instituto Mexicano del Seguro Social (IMSS) Metepec, Toluca City, and Medical Specialties Unit for the Detection and Diagnosis of Breast Cancer (UNEME-DEDICAM) of Health Institute of the State of Mexico (ISEM) Toluca City were collected, Appendix A.

#### Sample Treatment

Serum samples were obtained from healthy women (with no breast cancer or other pathology), patients with BC, and survivors of this disease. After that, a simple organic methanol-based extraction protocol was performed to obtain a dry lyophilized extract [42]. The extracts were reconstituted and derivatized by adding 50 µL of N-methyl-N-trimethylsilyl-trifluoroacetamide (MSTFA) and heated at 80 °C for 30 min in a dry block heater. After derivatization, an aliquot of 40 μL was transferred into a 200 μL vial insert and 10 μL of deuterated dicyclohexyl phthalate (2.5 ng/μL) were added as internal standards immediately prior to injection into a gas chromatograph coupled to a mass spectrometer (GC-MS). Quantitative analysis was performed using calibration curves with internal standards (Table 2).

### 2.3. Data Analysis

Chromatographic data were analyzed with Enhanced Data Analysis software (Agilent Technologies, Santa Clara, CA, USA). Peak identity was based on retention time and analyte standards and confirmed by mass spectrum. Internal standard calibration curves were used for quantitation.

#### 2.3.1. Reagents and Chemicals

Bisphenol-A, bisphenol-S, and MSTFA were purchased from Sigma-Aldrich (St. Louis, MO, USA). Phthalate diester standards, diethyl phthalate (DEP), di-n-butyl phthalate (DBP), butyl-benzyl phthalate (BBP), and bis-ethyl-hexyl phthalate (DEHP) were purchased from Chemservice Inc. (West Chester, PA, USA). Deuterated dicyclo-hexyl phthalate was from Accustandard. Methanol and ether were purchased from MilliporeSigma (Burlington, MA, USA).

#### 2.3.2. GC-MS Conditions

Analyses were performed in a gas chromatograph-mass spectrometer (7890-B/5977-B), Agilent Technologies, Santa Clara, CA, USA) with a quadrupole mass filter. Each sample was analyzed in duplicate. A 60 m DB-35 ms capillary column (250 μm × 0.25 μm) was used for chromatographic separation. High purity helium was used as the carrier gas at a 1.2 mL/min flow rate. The initial oven temperature was set to 80 °C for 1 min and then increased at 20 °C/min to 320 °C, with a 7 min hold. The mass spectrometer was operated in electronic ionization mode (70 eV) and in scan mode (25–430 Da). The temperatures were 300 °C for the transfer line, 230 °C for the ion source, and 150 °C for the quadrupole.

#### 2.3.3. Statistical Analysis

The statistical differences in the concentration of different compounds among the control group and patients with BC were determined by a non-parametric analysis, using the Mann–Whitney U test for paired comparisons. In addition, we compared the concentrations of the EDCs in controls, patients with BC, and survivors; for that purpose, we employed a one-way ANOVA, followed by a Kruskal–Wallis test. The specialized software package GraphPad Prism 6 version (San Diego, CA, USA) was used for the analysis. *p* < 0.05 was considered statistically significant.

## 3. Results

### 3.1. Comparison of Serum Levels of Phthalates and Bisphenols in Healthy Women between Mexico and Toluca City

The serum concentrations of different phthalates and bisphenols in control groups belonging to Mexico City (MX) and Toluca City (Toluca) are shown in Figure 1. We found significant differences in the serum levels of phthalates and bisphenols between the two cities. The level of DEP was higher in controls from Mexico City than in controls from Toluca City (Figure 1A). However, DBP (Figure 1B), DEHP (Figure 1C), and BBP (Figure 1D), were higher in Toluca than in Mexico City. Globally, we observed an increase of 2, 8, and 442 times for DBP, DEHP, and BBP, respectively. In contrast, we found a higher concentration of both BPA (Figure 1E) and BPS (Figure 1F) in the serum of controls in Mexico City.

### 3.2. Comparison of Serum Levels of Phthalates and Bisphenols in Patients with BC between Mexico and Toluca City

We also assessed the serum concentrations of different EDCs in patients with BC from both cities. We found a similar trend for DEP as found in the women without BC, (Figure 2A). Contrary to the observed data, DBP showed a significant increase in patients with BC belonging to Mexico City, which was 25 times higher than patients residing in Toluca City (Figure 2B). On the other hand, the serum concentrations of DEHP and BBP were higher in patients with BC from Toluca City, paralleling the results from healthy controls. Of note, the most significant difference between both cities in BC cases was in BBP levels, corresponding to 99 times greater in women with BC in Toluca City. Nevertheless, it is important to highlight that serum levels of DEHP and BBP in patients with BC belonging to Mexico City had notable increases (13 and 3.4 times more, respectively) when compared with healthy counterparts (Figure 2C versus Figure 2D).

Regarding BPA levels, there was no significant change between the two cities among patients with BC, which differed from the results in healthy controls (Figure 2E). We also observed that the diseased population from Toluca City did not show detectable serum levels of BPS, like the results in healthy controls (Figure 2F).

### 3.3. Comparison of Serum Levels of Phthalates and Bisphenols in Patients with BC Classified by Clinical Stage between Mexico and Toluca City

We found it important to group both the BC cases and controls from both cities and compare the levels of each compound to see the global behavior. The results showed only significant differences in the levels of DBP and BPS (Figure 3B,F).

Next, we decided to classify the data by clinical stage. In the case of patients from Mexico City, we only have 1 woman corresponding to the clinical stage I (Table 1), which made it difficult to compare the different serum levels of all of the compounds tested. Regarding patients with BC from Toluca City in stage I, we had 8 patients. Generally, DEP, DEHP, and BBP serum concentrations were found around 6000–8000 ng/mL; only the DBP concentration was around 400 ng/mL in this population. It is important to mention that no significant differences were found between the various phthalates tested (data not shown).

Regarding clinical stage II, DEP and DBP serum levels were significantly higher in patients with BC from Mexico City than in women with BC from Toluca City. While the concentrations of DEHP and BBP stayed significantly elevated in individuals with BC coming from Toluca City compared to patients from Mexico City in the same stage (data not shown), replicating the tendency described in Figure 2. Of note, the serum levels of DEHP and BBP in the population with BC belonging to Toluca City were increased compared with the cancer population from this city in stage I, reaching 10,000 ng/mL. For BPA, the patient from Mexico City and the patients from Toluca City had levels of BPA around 10 ng/mL. BPS was not detected in any women with BC at this stage, neither in Mexico City nor Toluca City (data not shown).

Like the combined results, no significant changes were observed in the serum levels of BPA of patients with stage II BC between the two cities (data not shown). As stated before, BPS was not detected in any patients coming from Toluca City.

Regarding the levels of phthalates in patients with BC in clinical stages III and IV we did not observe significant changes. in the serum levels of bisphenols and phthalates as were observed in stage II for both cities (data not shown). BBP concentration remained highly elevated in patients from Toluca City compared to those from Mexico City.

### 3.4. Comparison of Serum Levels of Phthalates and Bisphenols in Patients with BC by Molecular Type

Considering the widely known three different molecular subtypes of BC, we also analyzed the data based on this classification.

Regarding the patients with HER2+ and TN BC subtypes, we observed that DBP and BBP serum concentrations were significantly higher in these two subtypes than in the control group and patients with ER+ phenotype (Figure 4B,D). The contrary was observed with the concentrations of BPS, where women with ER+ BC disease showed higher serum concentrations of this bisphenol as compared with the other two subtypes, Figure 4F.

We also made the same classification for patients with BC residing in Toluca City. The data did not show any difference among the serum levels of the phthalates and bisphenols tested in the different subtypes of BC (data not shown).

### 3.5. Comparison of Serum Levels of Phthalates and Bisphenols in BC Survivors, Women after 5 Years Free of Recurrence, in Toluca City

We considered it important to compare if in the control, patients and patients who survived BC the levels of phthalates and bisphenols are affected in after their therapeutic treatment (Figure 5 and Table 3). According to the above, we obtained access to 19 samples of survivors of BC who are in the phase of survivorship care, which corresponds to a period of 5 or more years without the disease [43].

We observed significant differences in DEP (5A) DEHP (5B) and BBP (5D) when comparing the levels of controls and survival only from Toluca City. Of note, the levels of DEP were higher in survivals vs. controls (200 ng/mL, <100 ng/mL) (5A). We did not find significant differences in the levels of DBP (5B), while the levels of DEHP were significantly diminished (5C) and the levels of BBP (5D) were undetectable in survivals vs. controls, respectively. Notably, the levels of DEHP were diminished around 3.3 times in survivals vs. controls, both groups were only from Toluca City. Regarding bisphenols, there were no changes in BPA levels (5E) while the concentration of BPS was also undetectable in both controls and survivals (Figure 5).

In order to make these comparisons more precise, we made a global comparison among controls, patients with BC, and survivors of Mexico City and Toluca City. When BC patients were included, we only observed differentially significant downward changes in DEHP, DBP, and BBP levels in survivor women compared with controls and patients with BC (Table 3). It should be noted that the levels of BBP in survivals, controls, or patients with BC had always been high in the population belonging to Toluca City (data not shown). Impressively, these changes were abolished when comparing survivors with controls and patients with BC, since these levels were undetectable in the survivors population, Table 3

Regarding BPA and BPS, the results did not show significant changes in serum levels of BPA belonging to controls or survivors. In the case of BPS, we did not find any women residents of Toluca City (control or survivors) that had detectable levels of this compound (Figure 5F, Table 3).

Making broad comparisons, we found that DEP levels were significantly elevated in survivors compared to controls but unchanged in patients with BC.

For DBP there was a significant decrease in concentrations found in survivors compared to patients. With respect to DEHP, there were no differences between the groups, while for BBP the serum levels were significantly lowered in the survivors with respect to the control group and with respect to the patients with BC.

On the other hand, in patients with BC, BPA levels showed significant increases with respect to the control group and survivors, but there were no significant changes in BPS levels, Table 3.

## 4. Discussion

The major incidence of cases of BC is related to hormone dependence; however, in most cases, the causes of breast tumors are unknown, and molecular aberrations, environmental and genetic, and epigenetic factors may play a crucial role [44]. The purpose of this study was to quantify for the first time the serum levels of parent phthalates and bisphenols widely used in consumer products, including cosmetics, medications, food containers, etc., in Mexican women with or without BC and women who survived this disease after five years free of recurrence. The population enrolled for this study resided in two highly industrialized cities. Interestingly, we found that the concentrations of phthalates and bisphenols in the three groups of women evaluated have significant differences.

Both phthalates and bisphenols mimic hormone actions and may promote BC, carrying out several detrimental side effects at multiple levels [5,45] Preclinical evidence suggests that phthalates and bisphenols promote BC growth through ER signaling [13,46]. Nevertheless, the proliferation of estrogen-independent BC cells induced by phthalates and bisphenols has also been reported [11,47,48]. Existing epidemiologic studies on this topic are small and provide inconsistent results (Table 4). Our findings showed that all women residing in Mexico City had higher serum levels of DEP than in Toluca City. Of note, the levels of DEP seemed to be higher in patients with BC (>200 ng/mL) than in the control population (~100 ng/mL) in Mexico City. In contrast, the control and BC group of Toluca City showed similar DEP concentrations (~50 ng/mL). These results agree with the findings of the research group of López-Carrillo et al. (2010), who reported a significant positive association between the risk of BC with exposure to DEP in women residing in northern Mexico, as judged by the elevated urine concentrations of one of its metabolites, mono-ethyl phthalate (MEP) [20]. The above finding suggests that DEP is widely distributed not only in the northern part of Mexico but also in the country’s central zone. In addition, the exposure to DEP may be related to the increased incidence of BC in Latin American women.

On the other hand, our results about DBP exposure in patients with BC are consistent with a study of Danish patients [21]. The authors quantified the intake of phthalates in a medicine prescription in patients with BC, and they calculated the annual cumulative exposure to phthalates by capsule intake of cancer patients. The authors affirmed that high levels of DBP were associated with an approximately two-fold increase in the incidence rate of BC with an ER+ phenotype. It is worth noting that the effects of BBP and DEHP were not included in that study because these compounds were not used in Denmark [22].

Moreover, our study showed a remarkably significant increase in the serum concentrations of DBP in patients with BC in both cities, but especially in Mexico City, as compared with their corresponding control group (Figure 2B). When comparing the serum concentrations of DBP among the molecular subtypes of BC, it seemed that the HER2+ and TN types presented higher levels than the hormonal type. Interestingly, there were no significant changes between serum levels of DBP in controls and survivors of BC. This indicates that the increase in this compound could be related to the presence of malignant cells. However, further studies are needed to confirm this assumption.

Another compound related to a high BC incidence is DEHP. The urine concentrations of one of its metabolites, mono-ethyl-hexyl phthalate (MEHP), have been associated with BC progression in Alaskan native women [23]. Moreover, there is a positive association between urinary levels of DEHP metabolites, mainly mono(2-ethyl-5-carboxypentyl) phthalate (MECPP), and the risk of two associate reproductive neoplasms, BC and uterine leiomyoma [26]. These results are associated with the modulation of P53 and peroxisome proliferator-activated receptor-α (PPARα) signaling pathways [26]. Furthermore, DEHP can induce DNA damage in human cells affecting apoptosis and mitotic rate [49]. The results presented here do not show a significant change in DEHP levels in patients with BC compared to controls. However, women without BC from Toluca City displayed higher concentrations of DEHP than women from Mexico City. The above might reflect that in Toluca City most companies carry out activities derived from plastics and medical products. In addition, it is important to mention that the serum levels of DEHP significantly decreased in survivors of BC from this city. Considering that most of the survivors of BC enrolled in the present study presented a luminal subtype (Table 1), we might assume that the antihormone therapy that they received during their therapeutic regimen played an important role to achieve this effect, since DEHP was shown to have important estrogenic and progestogenic activity [12,50].

Our research also shows that there are increased serum levels of BBP in controls and BC cases in Toluca City compared to Mexico City residents, which are drastically diminished in survivors of BC. Our results disagree with the findings of López-Carrillo et al. (2010), who previously reported a significant negative association between the risk of BC and exposure to BBP in women with BC residing in northern Mexico [20]. There are few reports about the relationship between the increased risk of BC and the concentration of this phthalate. In fact, a report indicates that BBP and some other phthalates are not associated with increased BC risk [25]. However, the authors considered an important limitation when including postmenopausal women in their study since this period of life is not so relevant for the initiation of BC [22]. Therefore, we propose considering the following parameters: lifestyle, geographic region, metabolic parameters, polymorphisms, or enzyme alteration, and pre- and postmenopausal conditions, among others, in patients with BC to make more robust comparisons and assertions between the presence of phthalates and BC progression. Reeves et al. (2019) mentioned that the urinary concentration of phthalates, including BBP, did not increase the risk of developing invasive BC in postmenopausal women [22]. As such, we can suggest that perhaps the difference between the lack of BBP association that we observed, and the negative correlation observed by Lopez-Carrillo et al. (2010) is due to the geographic area. In addition, a deep study on the main products or industries that use BBP in their manufacture in these different regions should be considered to make stronger assumptions.

The interaction evoked by the co-exposure of flavonoids and phthalates, especially the BBP, can reduce the risk of BC [23]. Flavonoids could interact with several enzymes implicated in the metabolism of phthalates. The above agrees with other investigations that demonstrated pharmacological interactions when combining several EDCs with other agents resulting in toxic effects and alterations in the activity of certain enzymes [51,52,53]. In this regard, we visualize that all individuals are surrounded by different environmental pollutants that are taken in simultaneously, not separately, favoring their chronic exposure As was mentioned in the study of Mérida-Ortega et al. (2016), several compounds or the joint administration of drugs or foods can have an impact on some enzymes that metabolize phthalates and bisphenols [24]. Taking this notion into account, measuring the parent compounds and not their metabolites would reflect an individual’s exposure at a specific moment more accurately.

Regarding bisphenols, our results denoted typical concentrations of BPA found in women without BC, 5 and 20 ng/mL, in Toluca and Mexico City, respectively. The serum levels of this compound did not change significantly in women with BC. In accordance with our results, Yang et al. (2009) previously reported that they did not observe significant differences in the serum levels of BPA in Korean women with or without BC (control = 0.03 μg/L; cases = 0.61 μg/L) [40]. Similarly, there were no apparent changes in urinary levels of BPA in women with and without BC from New York City [28] Supporting this, Morgan et al. (2017) also reported no significant associations between BPA and BC [25]. In addition, another work by Trabert et al. (2014) also did not find a relationship between urinary levels of a glucuronide metabolite of BPA (BPA-G) and BC in women in the postmenopausal stage [27].

In addition, we observed that the levels of BPS were extremely low in women without and with BC. Some research groups have not found a strong relationship between high concentrations of BPA or BPS in human biological fluids and increased risk of BC, although several preclinical studies have demonstrated that low doses of BPA and BPS alter the proliferative cell dynamic of BC cells [14]. This suggests that low concentrations of bisphenols may determine the progression of BC.

With preclinical studies reporting the induced effects on BC cell proliferation and tumor growth by phthalates and bisphenols [12,14,54,55], the intratumoral quantification of them could be important to know the full extent of exposure and the body bioaccumulation rate, as these compounds can be stored in adipose tissue or brain. [56,57]. This becomes relevant since local drug delivery strategies with antitumor drugs are now being applied in therapeutics with promising results [58,59]. In addition, it is essential to highlight that both bisphenols and phthalates participate in the proliferation of BC but also, they can promote resistance to conventional therapy employed in this disease and decrease the effectiveness of anti-hormonal treatment used in BC and other neoplasms [60,61,62,63].

Regarding the role of phthalates and bisphenols as endocrine disrupters, different studies have demonstrated that these compounds perform distinct biological actions through their binding to nuclear and membrane steroid receptors, including ER, PR, androgen receptor (AR), proliferation activating peroxisome receptor, aryl hydrocarbon receptor (AhR), insulin receptor, the retinoic X receptor, glucocorticoid receptors, among others. In this manner, phthalates and bisphenols are responsible for triggering a plethora of genomic mechanisms [16,64].

In addition, in different models, EDCs can induce changes in enzymes with a crucial role in chromatin remodeling, such as DNA methyltransferases. An important example is the hypomethylation status of the promoter of nucleosome binding protein-1 and other vital molecules involved in cell differentiation, studied after exposure to different EDCs, including bisphenols and phthalates in different periods of life [65]. According to this knowledge, the participation of EDCs in the development of BC may be due to a chronic intake, not only to a critical exposition in a specific moment.

We consider it important to mention that although the Food and Drug Administration FDA and the European Medicines Agency EMA have established different permitted daily exposure of phthalates (≤0.01, and 4 mg/kg for DBP and DEP, respectively). These figures are not enough to give an idea of the blood concentrations that could be obtained from a daily exposure to these compounds. the same case for BPA (temporary oral TDI (TDI) of 4 µg/kg b.w./day) [66,67], and the low observed adverse effects level (LOAEL) of this compound is 50 mg/kg/day.

Regarding EDCs intake and exposure measurements, the daily intake of LOAEL concentrations of each compound has not been compared at the same time with their respective serum concentration. In addition, the metabolites are generally measured instead of the parental, unconjugated, non-metabolized forms of EDCs. However, previous studies have reported that the unconjugated serum levels of BPA in humans were measured and were found in a range from 0.2 to 20 ng/mL [68,69]. These serum values are comparable to what was found in this study. Nevertheless, the estimation of different EDCs in the general population can be based on the biological sample evaluated, daily intake by different routes, and populations, as mentioned in other studies [34,70]. However, as we previously discussed, the serum values of the parental compounds of different EDCs evaluated in the present study are like those previously reported by various research groups in other countries (Table 4). One of the limitations of the present study is the sample size, which is limited. A second limitation of this work is that we cannot compare the serum concentrations of environmental compounds with other works, as we measured the parental compounds and not the metabolites. However, the fact that serum levels of metabolites previously reported from different geographic regions are also in the order of ng/mL gives confidence regarding the accuracy of our data [71,72,73]. We consider that the results obtained in this study and the extensive literature previously published concerning the effects of phthalates and bisphenols and their counterproductive actions in BC point to the potential damage that can occur when exposed to these pollutants. We welcome the fact that in various countries and products, the use of these types of compounds by the industries has already been restricted. However, in Mexico, as in many places, we still do not have any government initiatives in which their use will be monitored.

## 5. Conclusions

This is the first study to quantify and compare the levels of serum concentrations of unaltered phthalates and bisphenols in Mexican women without and with BC as well as BC survivors. Phthalates were found in higher concentrations than bisphenols. Our results support the hypothesis that there is a positive association between exposure to phthalates and BC incidence. Additionally, serum concentration of phthalates significantly increases in different BC subtypes in Mexico City. DEP and DEHP concentrations are different between the BC and survivors groups. We consider that several governmental policies should be promoted to avoid or reduce the use and the consequent intake of environmental pollutants. On the other hand, considering that phthalates might promote resistance to cancer therapy, we propose that the evaluation of these pollutants could have clinical utility as a predictive tool in BC.

## Figures and Tables

**Figure 1 ijerph-19-08040-f001:**
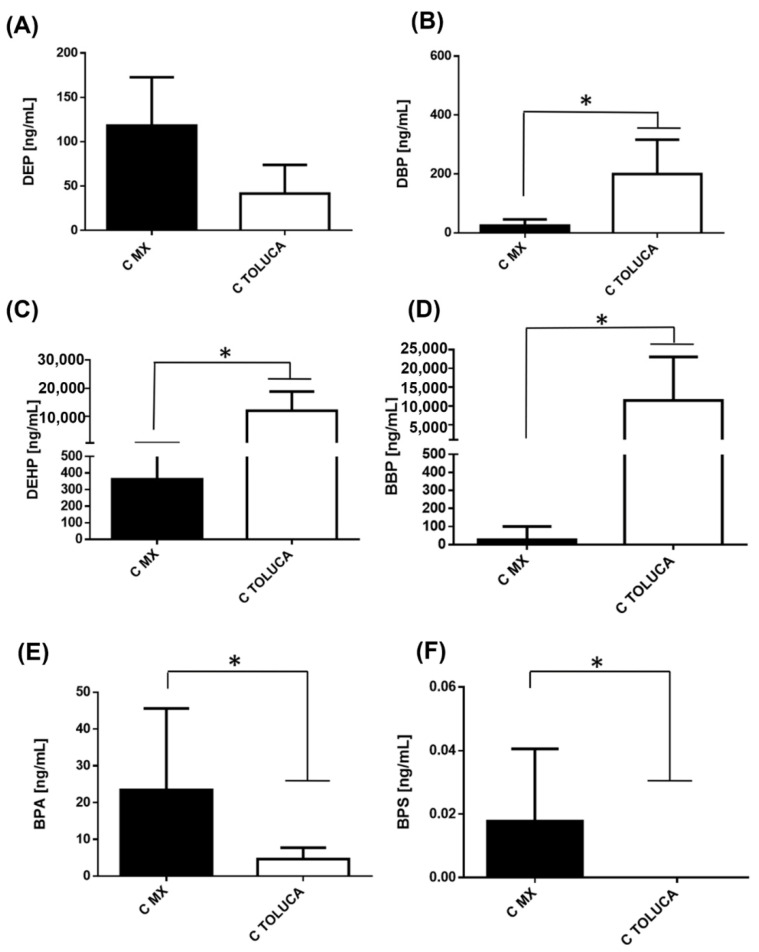
Levels of phthalates and bisphenols in women without BC Serum concentration levels of phthalates (**A**) DEP, (**B**) DBP, (**C**) DEHP, (**D**) BBP, (**E**) BPA, (**F**) BPS were measured in women (**C**) from Mexico City (black bar) and Toluca City (white bar). Bars represent the mean ± S.D. * *p* < 0.05 was considered statistically significant.

**Figure 2 ijerph-19-08040-f002:**
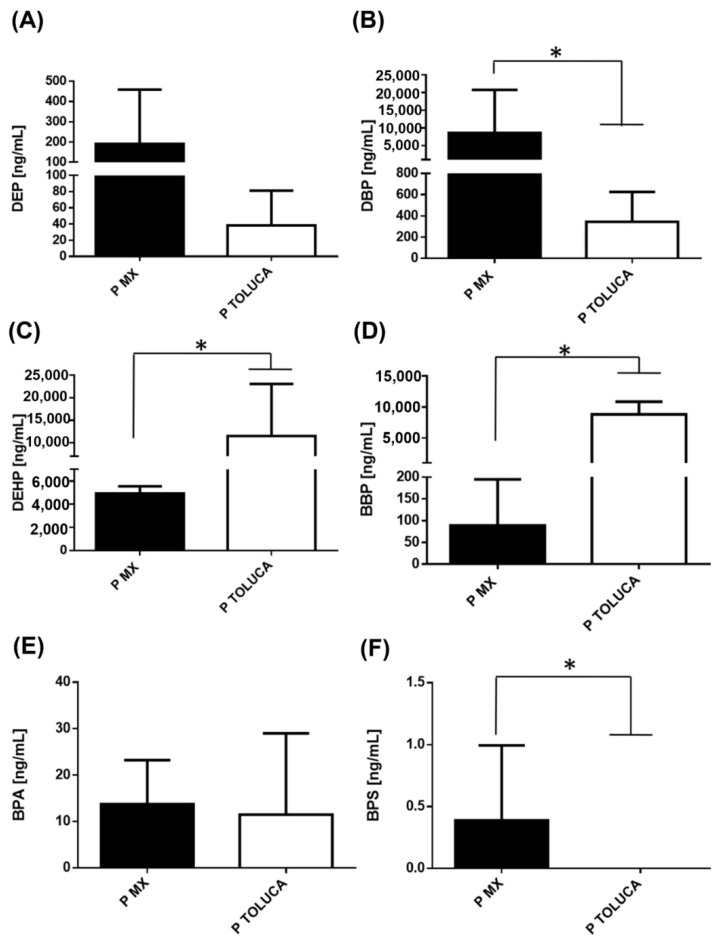
Levels of phthalates and bisphenols in BC cases. Serum concentration levels of phthalates (**A**) DEP, (**B**) DBP, (**C**) DEHP, (**D**) BBP, (**E**) BPA, and (**F**) BPS were measured in BC patients (P) from Mexico City (14 subjects/black bar) and Toluca City (32 subjects/white bar). Bars represent the mean ± S.D. * *p* < 0.05 was considered statistically significant.

**Figure 3 ijerph-19-08040-f003:**
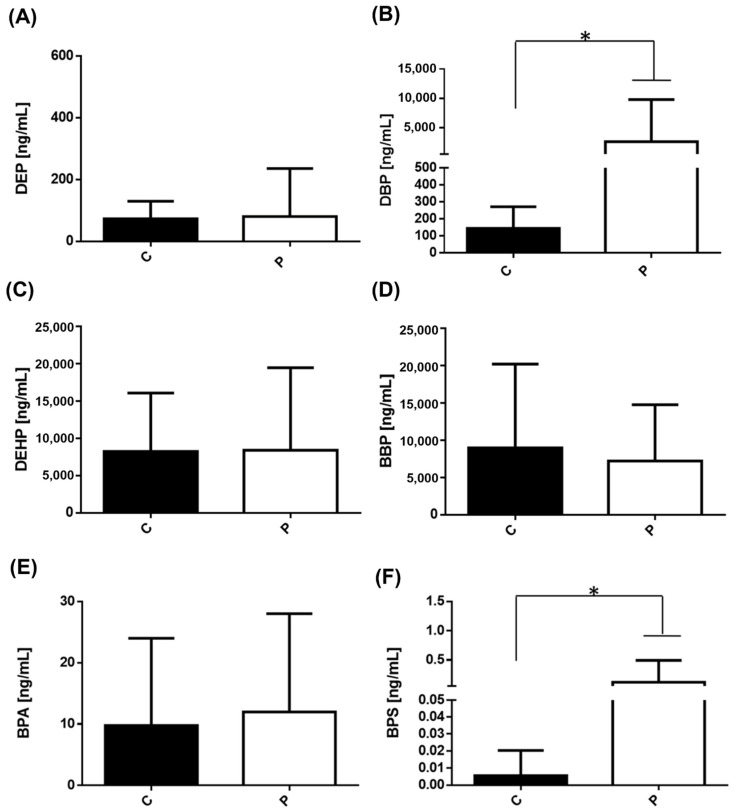
Levels of phthalates and bisphenols in controls and in patients with BC. Serum concentration levels of (**A**) DEP, (**B**) DBP, (**C**) DEHP, (**D**) BBP, (**E**) BPA, and (**F**) BPS were measured in controls (37 subjects/black bar) and in patients with BC (46 subject cases/white bar). Bars represent the mean ± S.D. * *p* < 0.05 was considered statistically significant.

**Figure 4 ijerph-19-08040-f004:**
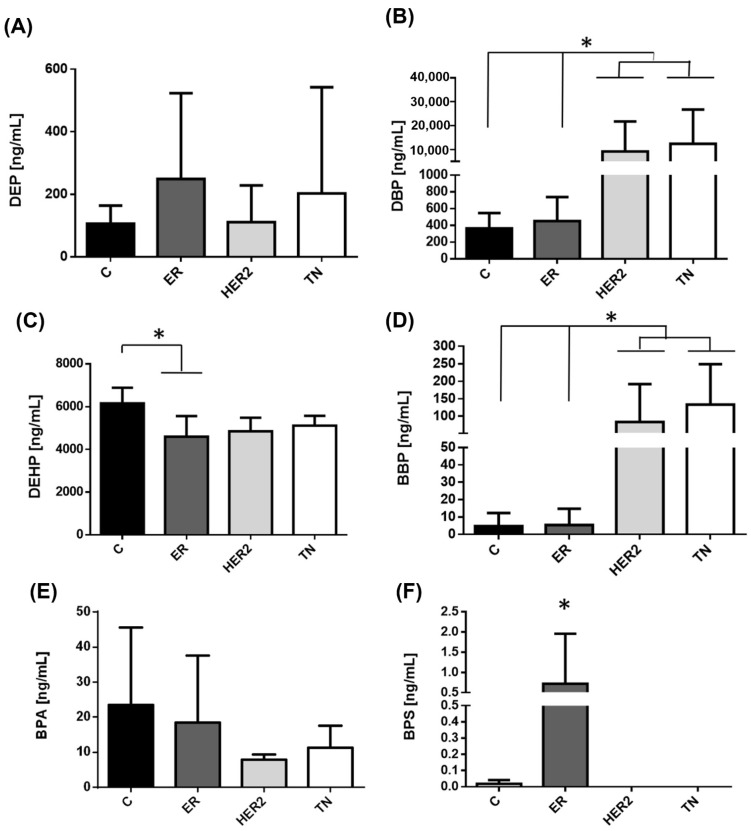
Levels of phthalates in patients with BC divided by molecular phenotype in Mexico City. Serum concentration levels of phthalates measured in different BC subtypes from Mexico City. The image displays the different serum concentrations of (**A**) DEP, (**B**) DBP, (**C**) DEHP, (**D**) BBP, (**E**) BPA, and (**F**) BPS in the control group (C/13 subjects, black bar) and compared with patients with BC divided by molecular phenotype; (ER+/4 subjects/dark gray bar; HER2+/4 subjects light gray bar, and TN/6 subjects, white bar). Bars represent the mean ± S.D. * *p* < 0.05 was considered statistically significant.

**Figure 5 ijerph-19-08040-f005:**
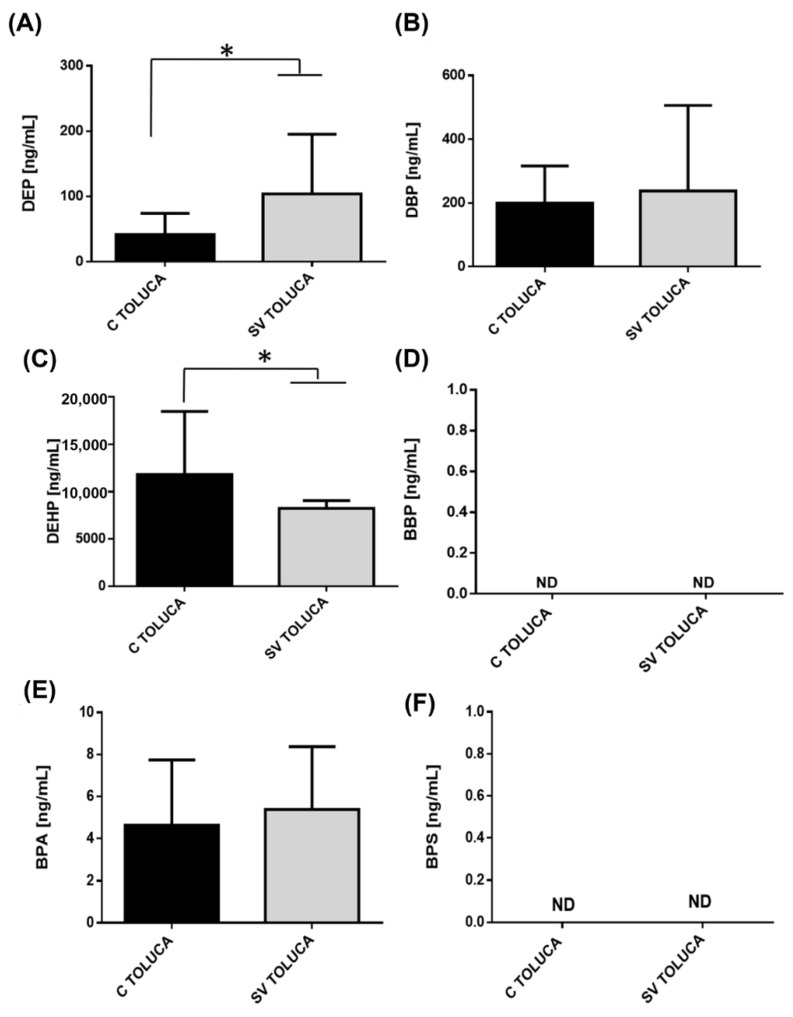
Levels of phthalates and bisphenols in survivors of BC in Toluca City. Serum concentration levels of phthalates measured in the Toluca City population among controls, and survivors. The image displays the different serum concentrations of (**A**) DEP, (**B**) DBP, (**C**) DEHP, (**D**) BBP, (**E**) BPA, and (**F**) BPS in the control group (black bar) and in women survivors of BC disease (gray bars). Bars represent the mean ± S.D. * *p* < 0.05 was considered statistically significant. ND = not detected.

**Table 1 ijerph-19-08040-t001:** Clinicopathological characteristics of patients with BC and survivors of this disease.

Patient	Age	Histological Classification	Molecular Classification	Stage
		Patients with BC from Mexico City		
1	60	Ductal	Luminal	IV
2	49	Ductal	TN	IV
3	42	Canicular	HER2	II
4	40	Ductal	TN	IV
5	46	Ductal	HER2	III
6	71	Canicular	Luminal	II
7	60	Lobulillar	TN	II
8	47	Canicular	TN	II
9	46	Ductal	Luminal	III
10	84	Ductal	TN	IV
11	37	Ductal	HER2	III
12	49	Ductal	Luminal	I
13	57	Ductal	TN	IV
14	48	Ductal	HER2	III
		Patients with BC from Toluca City		
A12	65	Canicular	Luminal	I
A17	58	Canicular	Luminal	0
A18	53	Canicular	Luminal	I
A19	52	Canicular	Luminal	II
A20	54	Canicular	TN	III
A21	48	Canicular	Luminal	III
A22	47	Canicular	Luminal	II
A24	53	Canicular	Luminal	II
A38	36	Canicular	TN	II
A41	49	Canicular	Luminal	I
A42	52	Canicular	NI	NI
A43	58	Canicular	TN	II
A44	64	Canicular	TN	II
A45	50	Canicular	Luminal	III
A46	58	Canicular	Luminal	I
A47	49	Canicular	HER2	IV
A48	62	Canicular	HER2	III
A49	43	Canicular	Luminal	0
A50	47	Canicular	Luminal	II
A51	46	Canicular	Luminal	I
A52	52	Canicular	NI	NI
A53	48	Canicular	NI	NI
A54	54	Canicular	HER2	II
A55	45	Lobulillar	TN	III
A57	45	Canicular	Luminal	II
A58	34	Canicular	Luminal	II
A59	40	Canicular	TN	IV
A60	52	Canicular	Luminal	0
A61	58	Canicular	Luminal	II
A62	48	Canicular	Luminal	II
A63	46	Canicular	HER2	III
A64	40	Canicular	Luminal	II
		BC survivors from Toluca City		
N24	44	Canicular	Luminal	II
N25	61	Canicular	Luminal	II
N26	46	Canicular	Luminal	III
N27	47	Canicular	Luminal	NI
N29	56	Canicular	Luminal	III
N30	56	NI	Canicular	NI
N31	43	NI	Luminal	III
N32	43	Canicular	Luminal	II
N33	56	Canicular	Luminal	0
N34	64	Canicular	HER2	0
N37	60	Canicular	Luminal	II
N40	49	Canicular	TN	II
N42	52	Canicular	Luminal	NI
N43	58	Canicular	Luminal	NI
N46	48	Canicular	HER2	0
N48	55	Canicular	Luminal	NI
N51	49	Mucinous	Luminal	NI
N52	53	Lobulillar	Luminal	NI
N53	52	Canicular	NI	NI

NI = non identified.

**Table 2 ijerph-19-08040-t002:** Quantitative analysis parameters.

Compound	Linear Range (pg)	Slope	Intercept	r^2^	Monitored Ions
BPA-TMS	0.1–10	0.0078	0.0375	0.9902	357, 358, 372
BPS-TMS	0.1–10	0.0025	0.004	0.9856	394, 379
DEP	10–10,000	0.002	0.2031	0.9948	149, 177, 76
DBP	10–10,000	0.0045	0.4721	0.9998	149, 205, 223
BBP	10–10,000	0.0019	0.0548	0.9999	149, 91, 206
DEHP	10–10,000	0.0027	0.2162	0.9995	149, 176, 279

**Table 3 ijerph-19-08040-t003:** Serum levels of EDCs in the 3 groups used to the study: controls, women with BC, and survivors of BC.

Compound	Controls	Women with BC	Survivors of BC	*p*-Value
DEP	72.70 ± 56.81	100.13 ± 179.27	103.77 ± 91.29 ^a^	0.0047
DBP	143.62 ± 126.91	3421.86 ± 8314.50 ^a^	237.55 ± 268.90 ^b^	<0.0001
DEHP	8259.79 ± 7835.16	8314.61 ± 12,975.97	8243.22 ± 821.73	0.3650
BBP	9226.21 ± 12,740.50	6425.55 ± 8805.81	0 ^a,b^	0.0284
BPA	4.63 ± 3.10	13.73 ± 17.09 ^a^	5.37 ± 2.99 ^b^	0.0254
BPS	0.01 ± 0.02	0.12 ± 0.37	0	0.1851

Data represent the average values plus/minus the standard deviation (SD) of each group; ^a^ = significant differences between controls, ^b^ = significant differences between women with BC.

**Table 4 ijerph-19-08040-t004:** Epidemiologic studies of measurement of phthalates and bisphenols in a patient BC.

Study Population	Objective	Results/Conclusion	Ref.
**Phthalates**			
Women with BC (233) residing in northern Mexico. 221 healthy women.	Phthalates were determined in urine samples by isotope dilution/high performance liquid chromatography coupled to tandem mass spectrometry.	Concentrations of mono-ethyl phthalate (MEP) were higher in cases (169.58 µg/g creatinine) than in controls (106.78 µg/g creatinine) Controls showed significantly higher concentrations of mono-n-butyl phthalate, mono(2-ethyl-5-oxohexyl) phthalate, and mono(3-carboxypropyl) phthalate (MCPP) than did the cases. The authors show for the first time that exposure to diethyl phthalate (DEP), the parent compound of MEP, may be associated with an increased risk of BC, whereas exposure to the parent phthalates of MBzP and MCPP (BBP and dioctyl phthalate, DOP, respectively) might be negatively associated with the incidence of this disease.	[20]
Association of phthalate exposures and breast cancer risk in a Danish nationwide cohort (1,122,042 women), using redeemed prescriptions for phthalate-containing drug products to measure exposure.	Drugs containing phthalates that were marketed in Denmark were registered in an internal Danish Medicines Agency database. The authors enrolled a Danish nationwide cohort of 1.12 million women at risk for a first cancer diagnosis on 1 January 2005. The authors calculated the annual cumulative phthalate exposure content of each filled prescription by multiplying the mass of phthalate per capsule by the fill amount. They calculated the cumulative milligrams of cellulose acetate (CAP) phthalate or polyvinyl acetate phthalate contained in all prescriptions filled by a patient during each year of follow-up. Finally, they made a multivariable Cox regression to estimate the associations between phthalate exposure and the incidence of invasive breast carcinoma according to the ER status of tumors.	High-level dibutyl phthalate (DBP) exposure (>10,000 cumulative mg) was associated with an approximately two-fold increase in the incidence rate of ER+ breast cancer. Cumulative exposures to DBP, cellulose acetate phthalate (CAP), and hydroxypropyl methylcellulose phthalate (HPMCP) were categorized as no exposure, 1 to 249 mg, 250 to 999 mg, 1000 to 9999 mg, and 10,000 mg or more. Cumulative exposure to DEP was categorized as unexposed, 1 to 9 mg, 10 to 99 mg, and 100 mg or more. Polyvinyl acetate phthalate (PVAP) exposure was rare and therefore modeled as a dichotomous variable (unexposed to any exposure [range, 1.3 to 682 cumulative grams]).	[21]
Case-control study within the Women’s Health Initiative (WHI) prospective cohort (419 invasive cases and 838 controls)	Quantification of 13 phthalate metabolites and creatinine in two or three urine samples per participant over one to three years	The urinary concentration of phthalates did not result in an increased risk of developing invasive BC in postmenopausal women. However, not entirely consistent, but the authors observed some positive effects between phthalate biomarker concentrations and BC diagnosed within three years. In addition, the majority of the positive association was significant for ER+/PR+ disease. It is worth noting that the positive effects estimated were closer to null and not statistically significant when analyses were extended to include case subjects diagnosed within five years or among the full study population. The above suggests that urine phthalate biomarker concentrations predict short-term, but not long-term BC risk. The authors grouped phthalate biomarkers by their parent phthalates by dividing the concentrations of each metabolite of a single parent by its molecular weight and then summing the concentrations across metabolites were corrected for creatinine concentration. The authors did not report a range of concentrations of phthalates, they reported the odds ratios established for each phthalate.	[22]
Case-control study of Alaskan native women 170 women (75 cases, 95 controls)	To measure the association between exposure to environmental chemicals and breast cancer. Seven to ten phthalate metabolites were measured in urine samples.	The authors found a potential association for DEHP exposure, which results in high levels of monoethylhexyl phthalate (MEHP) metabolites and the progression of breast cancer. The concentrations of phthalate metabolites were creatinine corrected (3.5 µg/g creatinine in controls and 5.3 µg/g creatinine in cases). Although urinary concentrations for most of the 7 phthalates metabolites were higher among cases than controls, these differences were not statistically significant. 40 of the 62 women who had invasive tumors had BC with an ER+/PR+ phenotype; the rest of them were negative for hormone markers. Urine concentrations were higher in 3 of 7 phthalate metabolites among women with ER-/PR-tumors. The differences were not statistically significant	[23]
Women resident of northern Mexico with histologically confirmed BC (233 individuals) and healthy controls (221 individuals)	To evaluate if phthalate exposure interacts with a flavonoid diet to promote the risk of BC. Urinary metabolites concentrations of nine phthalates were made and corrected to creatinine concentration.	A higher intake of anthocyanidins and flavan-3-ols (from vegetables), synergistically increased the a negative association between monobenzyl phthalate (MBzP), a metabolite of BBP, and the risk of BC. The consumption of some flavonoids may interact with exposure to phthalates with the risk of BC. Concentrations of urine metabolites of phthalates were found in a range of (5–139 μg/g creatinine).	[24]
U.S women with BC (cases = 43, controls = 1964 individuals)	Examination the relationship between the exposure to different endocrine disruptor compounds such as polychlorinated biphenyls (PCBs), BPA or phthalates; and risk of BC in U.S. women. The authors measured the urinary levels (ng/g) of BPA and ten metabolites of phthalates. Urinary levels were corrected to the creatinine concentration.	There were no significant associations between phthalates or BPA and BC.	[25]
Meta-analysis Search in the literature with different databases were performed in PubMed, Embase, and Cochrane library (2288 articles were included)	Assessment of the association between urinary phthalate metabolites and risk of BC and uterine leiomyoma.	DEHP metabolites were associated with an increased risk of BC as well as uterine leiomyoma. Important considerations for the authors: The ranges of cut-off levels of urinary phthalate metabolites were not consistent among the studies, which affect the general conclusions. Most of the studies reviewed analyzed the exposure of phthalates at a specific point in the disease, but it was impossible to calculate exactly the cumulative exposure dose, which might lead to limited conclusions. It is necessary to extend the study to different geographic regions or ethnic populations	[26]
**Bisphenols**			
Polish women with or without BC in post-menopause. (1962) BC cases (1338 postmenopausal) and 2241 controls (1529 postmenopausal)	To evaluate the association between urinary unconjugated BPA and a metabolite of BPA, the BPA-glucuronide (BPA-G) and risk of BC in postmenopausal women from Poland. Urinary levels of BPA-G were calculated and creatinine-adjusted (ng-BPa-G/mg creatinine). The range of BPA concentrations for BC cases was around 2.29–4.78 ng/mg; and in controls was 2.76 ng/mg	There was no association between BPA-G and women with BC in the postmenopausal stage.	[27]
711 women with BC and 598 women without BC belonged to Nassau and Suffolk Counties on Long Island, NY	To evaluate the association of seven urinary phenol biomarkers and BC incidence with subsequent mortality. The authors also examined the effect measure modification by body mass index (BMI). The measurements were made in urine samples and corrected for concentrations of creatinine	The concentrations of BPA found in the control group were around 1.2 μg/g creatinine and 1.3 in women with BC. There was no association between BPA levels and BC incidence. Mortality associations were more pronounced among women with high body mass index. BC incidences were strongly related to urinary concentrations of parabens	[28]
Korean women with or without BC (N = 167)	Potential associations between BPA exposure and risk of BC in Korean women among patients with BC and controls (N = 167). The levels of BPA were monitored in serum samples (μg/L)	There were no significant differences in the serum levels of BPA (conjugated + free form) between the cases and controls. The range of conjugated BPA levels was 0–13.87 μg/L, and the levels of free BPA were 0.012–0.04 μg/L.	[29]

## Data Availability

Data availability can be requested by writing to the corresponding author of this publication.

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
