# Peer review of "Association of Serum Levels of Plasticizers Compounds, Phthalates and Bisphenols, in Patients and Survivors of Breast Cancer: A Real Connection?"

_ijerph, 2022, doi:10.3390/ijerph19138040_

Round 1

Reviewer 1 Report

Comments

  1. The title has a term “plastic component” this may mislead the reader, “Association of serum levels of plastic components in patients 2and survivors of breast cancer: a real connection?” , if possible the term plastic components can be replaced with BPA or Phthalates or plasticizers etc.. It is my suggestion.
  2. The term endocrine disrupting chemical (EDCs) can be used in the keyword instead of environmental pollutant
  3. In the page no.30 line No 119, the term “. Healthy controls” is used, what are the criteria to assign healthy controls? Did their serum sample never shown any phthalates of other EDCs?
  4. Page 3, Line 124 – “included in this study 19 women” , the sample numbers could have been minimum 25 for better statistical analysis.
  5. Page 3, line 128, “On the other hand, the controls consisted of 13 (Mexico City) and 24 (Toluca)” why there is a difference in control numbers? Could have been equal numbers from both cities?
  6. Page 5, line 134, “accordance with the Declaration of 134 Helsinki,” did they conduct this study in Finland or Mexico?
  7. Page 6, line 178, “Comparison of serum levels of endocrine disruptor compounds” this term EDC could have been used in the subtract or title.
  8. In the results section 3.1, how will you represent two different subject numbers (13, 24) in a same bar diagram?? Same way in 3.2, (14, 32subjects in same bar)
  9. The results, discussion & conclusion could have been co-related well with the new finding and written more scientifically.

Author Response

  1. The title has a term “plastic component” this may mislead the reader, “Association of serum levels of plastic components in patients 2and survivors of breast cancer: a real connection?” , if possible the term plastic components can be replaced with BPA or Phthalates or plasticizers etc.. It is my suggestion.
  2. Both, phthalates and bisphenols are truly plastic components. The idea was to make our contribution sound to the general audience, so they can see the problem and danger with this compounds. However, we have modified the title accordingly to your suggestion
  3. The term endocrine disrupting chemical (EDCs) can be used in the keyword instead of environmental pollutant
  4. Thank you for your comment. It has been corrected as suggested
  5. In the page no.30 line No 119, the term “. Healthy controls” is used, what are the criteria to assign healthy controls? Did their serum sample never shown any phthalates of other EDCs?

  1. We meant to women that have not breast cancer, and. Based on questionnaire not apparent diseases. That it is why we grouped as healthy controls. However, for the sake of clarity the term was changed to “women without any declared pathology”. In fact our study shows that this group of women have also EDCs in their serums.
  2. Page 3, Line 124 – “included in this study 19 women” , the sample numbers could have been minimum 25 for better statistical analysis.
  3. We are sorry for the mistake. We only included this number of patients because they were the ones that attended to their regular medical consultation during the period of patient recruitment for our study. This sample recruitment was performed at convenience according to patient availability. Plus, since they were survivors, they did not showed any more during the duration time of the study. However, the statistical test used did allow us to perform it with 19 patients.
  4. Page 3, line 128, “On the other hand, the controls consisted of 13 (Mexico City) and 24 (Toluca)” why there is a difference in control numbers? Could have been equal numbers from both cities?
  5. Because of the period time for recruitment in both cities, it was not possible to obtain the same number with the same inclusion criteria. The medical consultation attendance is not the same in both cities.
  6. Page 5, line 134, “accordance with the Declaration of 134 Helsinki,” did they conduct this study in Finland or Mexico?

The study was conducted in Mexico City ; however, the declaration of Helsinki is taken as a world reference about the respect for the ethical principles of patients. In addition, it is a globally accepted compendium of considerations for human participants in various research studies, it does not necessarily indicate that a study was done in Finland [1]. However, the above, the study has the acceptance of ethical protocols of the respective institutions where the patient samples were taken, as well as where the samples were processed. But considering your suggestions, the phrase was eliminated from the text.  1.Puri, K.S., et al., Declaration of Helsinki, 2008: implications for stakeholders in research. J Postgrad Med, 2009. 55(2): p. 131-4.

  1. Page 6, line 178, “Comparison of serum levels of endocrine disruptor compounds” this term EDC could have been used in the subtract or title.
  2. It has been corrected as suggested
  3. In the results section 3.1, how will you represent two different subject numbers (13, 24) in a same bar diagram?? Same way in 3.2, (14, 32subjects in same bar)
  4. In both results sections 3.1 and 3.2, two different numbers of subjects are not being represented in the same bar, the black bars represent the subjects from Mexico City and the white bars the subjects from the City of Toluca.
  5. The results, discussion & conclusion could have been co-related well with the new finding and written more scientifically.
  6. Dear reviewer, we do not understand what you mean by a more scientific way. The results section describes what was found in each graph in detail. Regarding the discussion section, we include a table with the reports that have been made to-date, and we discuss point by point what was reported in our study and what was observed in previous studies. Finally, in conclusion, we are providing knowledge and support on the levels of phthalates mainly and their association with breast cancer, as well as the fact that various public health policies should be taken into account to avoid the use of these compounds. Our data support our previous studies in vivo models; this study aims to close the gap between in vivo and translational studies in patients. Please let us know if you still think it should be written differently.

Reviewer 2 Report

The study of “Association of serum levels of plastic components in patients and survivors of breast cancer: a real connection?” is interesting and is supposed to add great value to the association between EDCs exposure and incidence of breast cancer. However, there are many drawbacks in this study.

1.     As the authors have mentioned in the discussion, the sample size of the present study is too limited to get enough data to draw a conclusion.

2.     The study was not well designed to do the comparison analysis and the whole manuscript was not clearly organized.

3.     To take Abstract as an example, the abstract was not clearly summarized the study with the necessary information, such as how many patients were enrolled? What’s the main objective of this study? How to do the analysis to evaluate the association of exposure to phthalates or bisphenols and the incidence of breast cancer.

4.     In methods, the information of the patients lacked the location they came from, which really affected the understanding of the Results part.

5.     In the method of GC-MS, it’s necessary to address if the different internal standards were needed to test phthalates and bisphenols.

6.     For data analysis in this study, the authors mainly compared the levels of exposure between two cities. However, in order to figure out the association of compound exposure and BC, it is more important to compare the levels of exposure between controls and patients with BC. Also, regarding to the sub-group analyses, three analyses should be necessary: 1) The comparison of chemical exposure among the controls and the patients from different clinical stages should be done. 2) The comparison between controls and different molecular types should be stated as a main statement in 3.4. part. 3) In 3.5., the comparison among controls, patients with BC and survivors should be conducted.

7.     Also, the authors didn’t state their result clearly and logically. It’s hard to understand it, especially there are a lot of errors to match the text and figures in both result and discussion. For example, lines 208; in line 233, where were the 8 patients? In lines 240-242, the authors only said the increased changes but not showed if they were significant. In lines 261-263, no significant was shown in figure 4E, but the authors still stated the higher concentration of BPA in ER+ group than the other two subtypes. In lines 284-288, the figure didn’t match with the statement of authors.

I will not list all the drawbacks. This manuscript absolutely needs well organized especially for the data analysis.

Author Response

  1. As the authors have mentioned in the discussion, the sample size of the present study is too limited to get enough data to draw a conclusion.
  2. Dear reviewer, we are aware that the sample of our study is small and it is a flaw of the work, however, if we consider all the subjects involved, these add up to a total of 102 subjects, 37 controls, 46 cases of breast cancer and 19 survivals. In this sense, even with this sample number, we could see differences. It is evident that many more participants could be considered. We also want to highlight that this study was carried out during the past two years where the subject of covid and the obtaining of samples for the study of other diseases were very limited.
  3. The study was not well designed to do the comparison analysis and the whole manuscript was not clearly organized. To take Abstract as an example, the abstract was not clearly summarized the study with the necessary information, such as how many patients were enrolled? What’s the main objective of this study? How to do the analysis to evaluate the association of exposure to phthalates or bisphenols and the incidence of breast cancer.
  4. Your consideration has been taken into account
  5. In methods, the information of the patients lacked the location they came from, which really affected the understanding of the Results part.
  6. In the results section it is clearly stated where the samples were obtained: "Serum from different patients with BC was collected from Hospital 20 de Noviembre (Mexico City), and Unit of Medical Specialties for the Detection and Diagnosis of Breast Cancer of the National Center for Gender Equity and Reproductive Health (UNEME), Toluca City. Samples of survivors of BC were taken in the General Regional Hospital 251 of the Mexican Institute of Social Security (Toluca, State of Mexico)." Please specify what information is not clear in the results. In addition, a supplementary figure (flow chart) was included to facilitate the data understanding
  7. In the method of GC-MS, it’s necessary to address if the different internal standards were needed to test phthalates and bisphenols.
  8. It is stated in the Table 1. If what you meant is that, other standards to test for specificity were used, because of the method, always internal standards of every molecules measured were used.
  9. For data analysis in this study, the authors mainly compared the levels of exposure between two cities. However, in order to figure out the association of compound exposure and BC, it is more important to compare the levels of exposure between controls and patients with BC. Also, regarding to the sub-group analyses, three analyses should be necessary: 1) The comparison of chemical exposure among the controls and the patients from different clinical stages should be done. 2) The comparison between controls and different molecular types should be stated as a main statement in 3.4. part. 3) In 3.5., the comparison among controls, patients with BC and survivors should be conducted.
  10. The information you are requesting is located in figure 3, where the levels of different phthalates and bisphenols are compared in subjects without declared pathology (controls) and in patients with breast cancer in both cities. This graph clearly shows how levels of DBP between both control groups and patients are different, being higher in patients with breast cancer. Regarding comparisons by clinical stage, these are described in section 3.3. We did not observe significant differences between breast cancer patients in different clinical stages. In the section 3.4 we considered that the differences in serum levels of phthalates and bisphenols depending of different molecular types Finally, we made and include a table in order to see if reviewer does agree how it looks to compare the levels of the different phthalates and bisphenols in controls, patients and survivors.

Table

Compound

Controls

Women with BC

Survivors of BC

p value

DEP

72.70 ± 56.81

100.13 ± 179.27 b

103.77 ± 91.29 b

0.0047

DBP

143.62 ± 126.91 a

3421.86 ± 8314.50 a,b

237.55 ± 268.90 b

<0.0001

DEHP

8259.79 ± 7835.16

8314.61 ± 12975.97

8243.22 ± 821.73

0.3650

BBP

9226.21 ± 12740.5 a

6425.55 ± 8805.81 b

0 a,b

0.0284

BPA

4.63 ± 3.10 a

13.73 ± 17.09 a

5.37 ± 2.99

0.0254

BPS

0.01 ± 0.02

0.12 ± 0.37

0

0.1851

  1. Also, the authors didn’t state their result clearly and logically. It’s hard to understand it, especially there are a lot of errors to match the text and figures in both result and discussion. For example, lines 208; in line 233, where were the 8 patients? In lines 240-242, the authors only said the increased changes but not showed if they were significant. In lines 261-263, no significant was shown in figure 4E, but the authors still stated the higher concentration of BPA in ER+ group than the other two subtypes. In lines 284-288, the figure didn’t match with the statement of authors. I will not list all the drawbacks. This manuscript absolutely needs well organized especially for the data analysis.
  2. We do not agree with the reviewer comment. Taken into account that all other 3 reviewers did not found the manuscript as hard, and bad written as the reviewer stated, we are confident that the information provided it is clear. However, we have taken into account your comment and gone trough out the Ms and make specific changes, and, included new information.

Reviewer 3 Report

The paper 'Association of serum levels of plastic components in patients 2
and survivors of breast cancer: a real connection' describes the potential correlation between internal levels of two classes of plasticizers and breast cancer. The paper is clear and well written with minor English revisions. Indeed, some aspects need to be improved.

Introduction line 61: this ref (16) is related to BPA only. Please provide more appropriate reference for EDs

Introduction line 67: 'critical period of what? Please explain

Introducion lines 96-100: BPA has been banned from several consumer products (e.g. baby bottles, food-oriented packaging) in many countries due to its action as endocrine disrupting chemical. In February 2018, the EU introduced stricter limits on BPA in food contact materials. The sentence 'since this compound is associated with an increase of cell proliferation of
BC cells, different alternatives of structurally analogous compounds such as BPS have emerged' is not correct. Please modify.

Meterials and Methods,

statistical analysis: Has the number of healthy people and patients to be enrolled in the study checked for the appropriateness for statistical power?

3.1. Comparison of serum levels of endocrine disruptor compounds in healthy women between Mexico and Toluca City. This is not clear. The Authors talked about phthalates and bisphenols and not about 'endocrine disruptor  compounds'. Please explain or change the title. The same for the other sections.

3.3. Comparison of serum levels of endocrine disruptor compounds....... this title is unclear and it doesn't correspond to the Table legend shown below.

The classification of the data by clinical stage should be described in a separate Table for clarity

The senteces from line 437 to 445 of the Discussion are meaningless. For every chemicals, the No/Low Observed Adverse Effect Levels (NOAEL/LOAEL) are derived from toxicological studies and the most sensitive NOAEL/LOAEL is used to define specific safety levels. It is completely uclear why the Authors cited 'some' NOAEL/LOAEL and not the corresponding TDI/ADI. As an example,   daily exposure to specific phthalates used in orally administered preparations. Proposed limits for DEP and DBP exposure have been set to 4.0 and 0.01 mg/kg/day, respectively by EMA. A less restrictive limit for DBP exposure has been set to 0.1 mg/kg/day by FDA.

Reff 69-70: as an example, more recent data on Italian BPA urinary concentrations are available on Tait et al Int J Environ Res Public Health. 2021 Nov 12;18(22):11846. doi: 10.3390/ijerph182211846

Author Response

  1. The paper 'Association of serum levels of plastic components in patients 2
    and survivors of breast cancer: a real connection' describes the potential correlation between internal levels of two classes of plasticizers and breast cancer. The paper is clear and well written with minor English revisions. Indeed, some aspects need to be improved.
  2. Thank you, we appreciate your comment
  3. Introduction line 61: this ref (16) is related to BPA only. Please provide more appropriate reference for Eds
  4. It has been corrected as suggested
  5. Introduction line 67: 'critical period of what? Please explain
  6. We meant critical periods of development. It has been included in the text.
  7. Introducion lines 96-100: BPA has been banned from several consumer products (e.g. baby bottles, food-oriented packaging) in many countries due to its action as endocrine disrupting chemical. In February 2018, the EU introduced stricter limits on BPA in food contact materials. The sentence 'since this compound is associated with an increase of cell proliferation of
    BC cells, different alternatives of structurally analogous compounds such as BPS have emerged' is not correct. Please modify.
  8. Thank you for pointing out to this. However, I am not sure I understood. On the one side, there has been more restrictions on the use of BPA, fact that lead to other alternatives of plasticizers, such as BPS, BPD, BPF. Which are structurally similar. And, also it is truth that it has been demonstrated that BPA and BPS have an effect increasing BC cells proliferation. However, we did change the sentence a little so it may be more precise.

Meterials and Methods,

  1. statistical analysis: Has the number of healthy people and patients to be enrolled in the study checked for the appropriateness for statistical power?
  2. We consider that statistical tests are appropriate, in general we perform non-parametric analysis, using the Mann–Whitney U test for paired comparisons with a 95% confidence interval. In addition, we included a one way ANOVA followed by a Kruskal-Wallis test to compare differences among controls, patiets with BC and survivals.. We also consider a p < 0.05 value as a significant difference. If you consider that it is incorrect, please suggest a specific statistical analysis.
  3. Comparison of serum levels of endocrine disruptor compoundsin healthy women between Mexico and Toluca City. This is not clear. The Authors talked about phthalates and bisphenols and not about 'endocrine disruptor  compounds'. Please explain or change the title. The same for the other sections.
  4. Your consideration has been taken into account and it has been corrected as suggested.

  1. Comparison of serum levels of endocrine disruptor compounds....... this title is unclear and it doesn't correspond to the Table legend shown below.
  2. It has been corrected as suggested.
  3. The classification of the data by clinical stage should be described in a separate Table for clarity
  4. We did not observe significant differences between stages. As was mentioned in the text significant differences were observed in the serum levels of DBP and BBP in patients with breast cancer only at the clinical stage II. So we believe that including a table with data that were not significant would be repetitive (Figure 3).
  5. The senteces from line 437 to 445 of the Discussion are meaningless. For every chemicals, the No/Low Observed Adverse Effect Levels (NOAEL/LOAEL) are derived from toxicological studies and the most sensitive NOAEL/LOAEL is used to define specific safety levels. It is completely unclear why the Authors cited 'some' NOAEL/LOAEL and not the corresponding TDI/ADI. As an example,   daily exposure to specific phthalates used in orally administered preparations. Proposed limits for DEP and DBP exposure have been set to 4.0 and 0.01 mg/kg/day, respectively by EMA. A less restrictive limit for DBP exposure has been set to 0.1 mg/kg/day by FDA. Reff 69-70: as an example, more recent data on Italian BPA urinary concentrations are available on Tait et al Int J Environ Res Public Health. 2021 Nov 12;18(22):11846. doi: 10.3390/ijerph182211846
  6. A clarifying sentence have been added mentioning TDI instead of NOAEL/LOAEL. However, it is unclear why reviewer wants the TDI, since for reference to public policies, both NOAEL/LOAEL are used. In the cited article, the levels are in urine, and, not the parental compounds, but the metabolites. For years that has been the gold standard. However, the most important point of this communication it is that we found parental levels of compounds, and, that they are in serum. This talks about the degree of exposure (chronic) that we may be, since all of those compounds may be in the air (we breath) in the water (we drink) and in the food (we eat), and are also present in many daily products that we use (deodorants, perfumes, make up). Thus, the measure of metabolites may be ok, but it is not as accurate as measuring parental compounds in serum.

Reviewer 4 Report

The manuscript written by Segovia-Mednoza et al concerns on the problem of influence of endocrine disruptors on the development of breast cancer. The problem has high priority according to the epidemiological data.

In my opinion the articel is very valuable and should be publicated, however it demands some minor corrections.

Introdution:

Please comment also the effect of product of degradation of endocrine disruptors (eg. isopropylphenol forming in the Fenton reaction from BPA).

It could be valuable if the authors add some information about in silico studies of endocrine disruptors (see eg. Makarova, Katerina, et al. "Screening of toxic effects of bisphenol A and products of its degradation: zebrafish (Danio rerio) embryo test and molecular docking." Zebrafish 13.5 (2016): 466-474.)

Results:

In my opinion it would be more clear if the results are collected in one table instead of series of charts. It is much easier to compare numerical values.

Discussion:

-line 315 - Of note, the levels of this compound seemed to be higher... - please be more precise, scientific, it is too generally.

- please correct the citations, lack of brackets in the text (line 303, 364, 369, 375)

- table 3, the 1st study - section results, the concentrations are 169,58 ug/g and 106,780ug/g. Why different accuracy? (0,01ug vs 0,001ug)

References:

Please, delete duplicated numbers from 7 to the end,

Author Response

  1. The manuscript written by Segovia-Mednoza et al concerns on the problem of influence of endocrine disruptors on the development of breast cancer. The problem has high priority according to the epidemiological data. In my opinion the articel is very valuable and should be publicated, however it demands some minor corrections.
  2. Thank you for your comments.
  3. Introduction. Please comment also the effect of product of degradation of endocrine disruptors (eg. isopropylphenol forming in the Fenton reaction from BPA).
  4. We have included a sentence about the Fenton effect in the introduction, considering your suggestion.
  5. It could be valuable if the authors add some information about in silico studies of endocrine disruptors (see eg. Makarova, Katerina, et al. "Screening of toxic effects of bisphenol A and products of its degradation: zebrafish (Danio rerio) embryo test and molecular docking." Zebrafish 13.5 (2016): 466-474.)
  6. We have included a sentence about that fact in the introduction, considering your suggestion
  7. Results: In my opinion it would be more clear if the results are collected in one table instead of series of charts. It is much easier to compare numerical values.
  8. We appreciate your comment, however, we consider that the fact of representing the results in one large table would be not as easy to follow and read for the reader, in addition to highlighting the significant differences between groups would also be complicated. The total the samples are 102, and a large Table would be very difficult to follow.
  9. Discussion: -line 315 - Of note, the levels of this compound seemed to be higher... - please be more precise, scientific, it is too generally.
  10. We have been more precise, and the numerical value were added.
  11. please correct the citations, lack of brackets in the text (line 303, 364, 369, 375)
  12. It has been corrected as suggested
  13. table 3, the 1st study - section results, the concentrations are 169,58 ug/g and 106,780 ug/g. Why different accuracy? (0,01ug vs 0,001ug)
  14. We have corrected the Table. Indeed, we stick to the 0.01 scale.
  15. References: Please, delete duplicated numbers from 7 to the end,
  16. Duplicate numbers have already been removed.

Round 2

Reviewer 2 Report

The authors have addressed most of the questions and have done the necessary revisions. The manuscript is much better than the last draft. There are still some drawbacks as follows.

1.     In line 38 of abstract, “DBP showed---?” 

2.     In methods, the information of the patients still lacked the location they came from, which really affected the understanding of the Results part. How many patients were from Mexico or Toluca city? The same information was missed in table 1. Cannot recognize the location for the patients in the list. At least, it’s necessary to add note under the table to show their locations. For example, “patients 1-14 were from---; A12-A64 were from---: N24-N53 were from---” Also, cannot find supplementary figure. 

3.     In the legend of Figure 3, there is an error in the number of patients of BC. It should be 46 but not 48.

4.     Cannot find Figure 5. Accidentally deleted?

5.     In Table 3, please add note to show what “a” and “b” mean?

Author Response

The authors have addressed most of the questions and have done the necessary revisions. The manuscript is much better than the last draft. There are still some drawbacks as follows.

Q: In line 38 of abstract, “DBP showed---?” 

A: Thank you for pointing this out. It was a finger error; it has been corrected as suggested.

 Q:  In methods, the information of the patients still lacked the location they came from, which really affected the understanding of the Results part. How many patients were from Mexico or Toluca city? The same information was missed in table 1. Cannot recognize the location for the patients in the list. At least, it’s necessary to add note under the table to show their locations. For example, “patients 1-14 were from---; A12-A64 were from---: N24-N53 were from---” Also, cannot find supplementary figure. 

A: The identification of the patients was already added in Table 1.

Q: In the legend of Figure 3, there is an error in the number of patients of BC. It should be 46 but not 48.

A: We thank you for your input. It has been corrected as suggested

Q: Cannot find Figure 5. Accidentally deleted?

A: Indeed. We are sorry for the mistake. The figure was included again now

Q: In Table 3, please add note to show what “a” and “b” mean?

  1. It has been corrected. The significance of the letters a and b has already been included.

We thank reviewer for her/his hard work and excellent comments in reviewing our manuscript. As a result, a much better draft of the same was generated. We hope that our responses will satisfy reviewer.